# ZRF1 is a novel S6 kinase substrate that drives the senescence programme

Manuela Barilari[1,2,3,†], Gregory Bonfils[1,2,3,†], Caroline Treins[1,2,3,†], Vonda Koka[1,2,3], Delphine De Villeneuve[1,2,3], Sylvie Fabrega[4] & Mario Pende[1,2,3,*] (ID)

## Abstract

The inactivation of S6 kinases mimics several aspects of caloric restriction, including small body size, increased insulin sensitivity and longevity. However, the impact of S6 kinase activity on cellular senescence remains to be established. Here, we show that the constitutive activation of mammalian target of rapamycin complex 1 (mTORC1) by tuberous sclerosis complex (TSC) mutations induces a premature senescence programme in fibroblasts that relies on S6 kinases. To determine novel molecular targets linking S6 kinase activation to the control of senescence, we set up a chemical genetic screen, leading to the identification of the nuclear epigenetic factor ZRF1 (also known as DNAJC2, MIDA1, Mpp11). S6 kinases phosphorylate ZRF1 on Ser47 in cultured cells and in mammalian tissues in vivo. Knock-down of ZRF1 or expression of a phosphorylation mutant is sufficient to blunt the S6 kinase-dependent senescence programme. This is traced by a sharp alteration in p16 levels, the cell cycle inhibitor and a master regulator of senescence. Our findings reveal a mechanism by which nutrient sensing pathways impact on cell senescence through the activation of mTORC1-S6 kinases and the phosphorylation of ZRF1.

**Keywords** mTOR; senescence; S6 kinase; tuberous sclerosis complex; ZRF1
**Subject Categories** Ageing; Autophagy & Cell Death; Signal Transduction
The EMBO Journal (2017) 36: 736–750

## Introduction

Caloric restriction has lifespan benefits across different species, from unicellular organisms such as yeast to small primates and possibly humans (Longo et al, 2015). Among the intracellular signal transduction pathways adapting growth, metabolism and ageing to nutritional cues, the target of rapamycin complex 1 (TORC1) has a central role (Laplante & Sabatini, 2012). The serine/threonine kinase activity of TOR in TORC1 is upregulated by a large variety of anabolic signals, including amino acids, glucose, lipids, growth factor peptides, mitochondrial metabolites, oxygen and energy supplies. Genetic alterations of TORC1 elements affect ageing in yeast, flies, worms and mice (Lamming et al, 2013). The allosteric TOR inhibitor rapamycin has been reported to be the first longevity drug acting in mammals, that is, in genetically heterogeneous mice (Harrison et al, 2009). However, both caloric restriction and rapamycin treatments have some adverse effects on organismal physiology (Wilkinson et al, 2012). Deciphering the molecular targets downstream of mammalian TOR (mTOR) that specifically affect age-related disorders may help the design of safer interventions.

S6 kinases 1 and 2 (S6Ks) are mTORC1 substrates that in turn possess serine/threonine kinase enzymatic activity (Dann et al, 2007). S6K1 and S6K2 are homologous proteins sharing similar modes of regulation and substrate specificities, which have been mainly inferred from the study of S6K1 (Pende et al, 2004). mTOR phosphorylates Thr-389 of S6K1 in the regulatory T loop, an event required for S6K1 activation. Among the known mTORC1 substrates, S6K1 phosphorylation requires a relatively high mTOR-specific activity (Kang et al, 2013). Hence, S6K1 phosphorylation is extremely sensitive to rapamycin treatment or nutrient levels, while the phosphorylation of other mTORC1 substrates is modulated to a lower extent. The modulation of S6K1 activity is therefore a sensitive functional adaptation to nutrient availability in the cell. Interestingly, genetic studies have revealed that S6K1 deletion in mice phenocopies a number of physiological adaptations to dietary restriction. Although S6K1-deficient mice have normal food intake, their body weight is reduced with a defect in cell size being particularly evident in metabolic tissues such as fat, skeletal muscle and pancreatic beta cells (Pende et al, 2000; Um et al, 2004; Ohanna et al, 2005; Aguilar et al, 2007). The insulin levels in their blood are low, while insulin sensitivity in peripheral tissues is increased. S6K1 mutant mice do not become obese when fed a high fat diet. Finally, their lifespan is longer as compared to control mice (Selman et al, 2009). Taken together, these findings suggest a linear pathway from nutrient availability to the control of growth and ageing through the activation of mTORC1 and S6K1.

---

1  Institut Necker-Enfants Malades, Paris, France
2  Inserm, U1151, Paris, France
3  Université Paris Descartes, Sorbonne Paris Cité, Paris, France
4  Plateforme Vecteurs Viraux et Transfert de Gènes, IFR94, Hôpital Necker Enfants-Malades, Paris, France
   *Corresponding author. Tel: +33 1 72 60 63 86; Fax: +33 1 72 60 64 01; E-mail: mario.pende@inserm.fr
   †These authors contributed equally to this work

  

Senescence is an irreversible cell cycle arrest in the G1 phase (Munoz-Espin & Serrano, 2014). Although it can also be observed during embryonic development and after oncogenic insults, senescence increases during ageing (Lopez-Otin *et al*, 2013). In primary cell cultures, entry into senescence usually requires proliferation in the presence of DNA damage and is controlled by the tumour suppressors p53, retinoblastoma protein (Rb) and the gene products of the Ink4a/ARF locus, p16 and Arf (Munoz-Espin & Serrano, 2014). Consistent with its known role on growth and ageing, mTORC1 activity promotes senescence. Mouse embryonic fibroblasts (MEFs) with a constitutive activation of mTORC1, due to loss-of-function mutations in upstream negative regulators TSC1, TSC2 or PTEN (phosphatase and tensin homolog), senesce faster than wild type, and this effect is blocked by rapamycin (Zhang *et al*, 2003; Alimonti *et al*, 2010). Although the sensitivity to rapamycin and to the S6K1 pharmacological inhibitor PF-4708671 suggests an involvement of S6K1 in this process (Leontieva *et al*, 2013), whether and how S6 kinases control senescence remain to be established.

The known S6K substrates belong to four broad classes (Dann *et al*, 2007; Ma & Blenis, 2009). The largest class are proteins involved in the RNA metabolism and protein synthesis: ribosomal protein S6 (rpS6), eukaryotic initiation factor 4B (eIF4B), PDCD4 (programmed cell death protein 4), eukaryotic elongation factor 2 kinase (eEF2K), carbamoyl-phosphate synthetase 2, aspartate transcarbamylase and dihydroorotase (CAD) and the β-subunit of chaperonin containing TCP1 (CCTβ). Second group are proteins involved in the retrograde control of insulin and mTOR signalling: insulin receptor substrate 1 (IRS1), mTOR, rictor (rapamycin insensitive companion of mTOR), Sin1 (stress-activated map kinase-interacting protein 1). Third are proteins involved in metabolic reprogramming: peroxisome proliferator-activated receptor gamma coactivator 1-alpha (PGC1α), glycogen synthase kinase-3-α and glycogen synthase kinase-3-β (GSK3α and β), AMP-activated protein kinase catalytic subunit-α (AMPKα). The fourth class are proteins with oncogenic or tumour suppressor activities: Bcl2-associated agonist of cell death (BAD), glioma-associated oncogene 1 (Gli1) and murine double-minute 2 protein (Mdm2). The molecular programme responsible for the age-related decline and senescence is still unclear.

To determine the functional contribution of the S6 kinase signalling in cellular senescence control, we evaluated the impact of S6 kinases deletion in TSC1 mutant MEFs, which undergo premature senescence. We demonstrate that S6 kinases are required for the cellular senescence programme triggered by TSC1 mutation. By performing a chemical genetic screen to search for novel S6 kinase substrates, we identify ZRF1 as a molecular target relevant to this process.

## Results

TSC1 and TSC2 deletion in MEFs is known to cause premature senescence (Zhang *et al*, 2003). To dissect the functional contribution of S6K activity, we scored the effects of the combined deletion of TSC1 and S6K1/2 on MEF senescence. In primary MEFs after adenoviral Cre-mediated deletion of the TSC1-floxed allele, increased rpS6 phosphorylation was abrogated by the concomitant deletion of S6K1/2 (Fig 1A). As expected, Akt phosphorylation inversely correlated with S6K activity, due to the negative feedback loop of mTORC1/S6K on mTORC2/Akt (Um *et al*, 2004; Hsu *et al*, 2011; Yu *et al*, 2011). Consistent with previous studies, TSC1-deficient cells, after adenoviral Cre transduction, underwent early senescence, as assessed by senescence-associated (SA) β-galactosidase staining (Fig 1B) and population doubling time (Fig 1C). The senescence programme was accompanied by upregulation of cell cycle arrest and senescence markers. These included p53 and its transcriptional target, the cell cycle inhibitor p21, together with p53-independent responses, such as the upregulation of the cell cycle inhibitor p16 (Fig 1A). As expected, the regulation of p16 and p21 abundance was at the mRNA level, while p53 was controlled post-transcriptionally (Fig 1D). Strikingly, S6K1/2 deletion blunted the senescence response (Fig 1B and C). At the molecular level, S6K1/2 deletion selectively affected p16 levels while the p53 response was spared (Fig 1A and D).

Since oncogene-induced senescence may be a consequence of hyperproliferation and DNA damage (Munoz-Espin & Serrano, 2014), the levels of DNA damage response mediators, phosphorylated histone H2AX (γ-H2AX) and 53BP1 foci, were measured by immunofluorescence at late passage P7 (Fig 2A and B). The deletions of TSC1 and S6K1/2 were not associated with these markers of DNA damage. In addition, S6K deletion at early passages P1 and P2 did not inhibit cell proliferation [data not shown and (Pende *et al*, 2004)]. Taken together, our data demonstrate that S6Ks are specifically required for the p16 branch of the senescence programme downstream of mTORC1 activation, without causing DNA damage and hyperproliferation. Interestingly, the overexpression of p16 has been shown to induce senescence in the absence of DNA damage (Coppe *et al*, 2011).

**Figure 1. S6 kinases control p16 expression and the senescence response in TSC1 mutant cells.**

A  Immunoblot analysis of *Tsc1*^fl/fl^ and *Tsc1*^fl/fl^ *S6K1*^−/−^*S6K2*^−/−^ primary MEFs transduced with GFP or GFP-Cre adenovirus and harvested from 9 to 12 days post-infection (passage P6 or P7). Proteins were analysed by Western blot using the indicated antibodies. Quantification by densitometric analyses of p53, p21 and p16 protein levels is presented as a graph. Data are normalized to β-actin and expressed as a fold change relative to the control cells. Data are presented as mean ± SD of at least three independent experiments. *P < 0.05, **P < 0.01, ANOVA multiple comparisons.

B  Bright-field images of β-galactosidase staining performed on *Tsc1*^fl/fl^ and *Tsc1*^fl/fl^ *S6K1*^−/−^*S6K2*^−/−^ primary MEFs, transduced with GFP or GFP-Cre adenovirus at passage P3 and fixed 12 days post-infection, at passage P7. Scale bar: 50 μm. Data are presented in a graph as mean ± SEM of at least three independent experiments; ****P < 0.0001, n.s. not significant, ANOVA multiple comparisons.

C  Population doubling analysis of *Tsc1*^fl/fl^ and *Tsc1*^fl/fl^ *S6K1*^−/−^*S6K2*^−/−^ primary MEFs transduced at P1 with GFP or GFP-Cre adenoviruses was determined for nine passages. Data are presented in the curve as mean ± SEM of at least three independent experiments; *P < 0.05, multiple t-tests.

D  Quantitative RT–PCR detection of the level of expression of the indicated genes in *Tsc1*^fl/fl^ and *Tsc1*^fl/fl^ *S6K1*^−/−^*S6K2*^−/−^ primary MEFs transduced at P3 with GFP or GFP-Cre adenovirus and collected at passage P7. Expression levels are corrected for expression of the control gene (β-actin) and presented in the graph as fold changes relative to control cells. The values plotted are means ± SEM of three independent experiments; *P < 0.05, multiple t-tests.

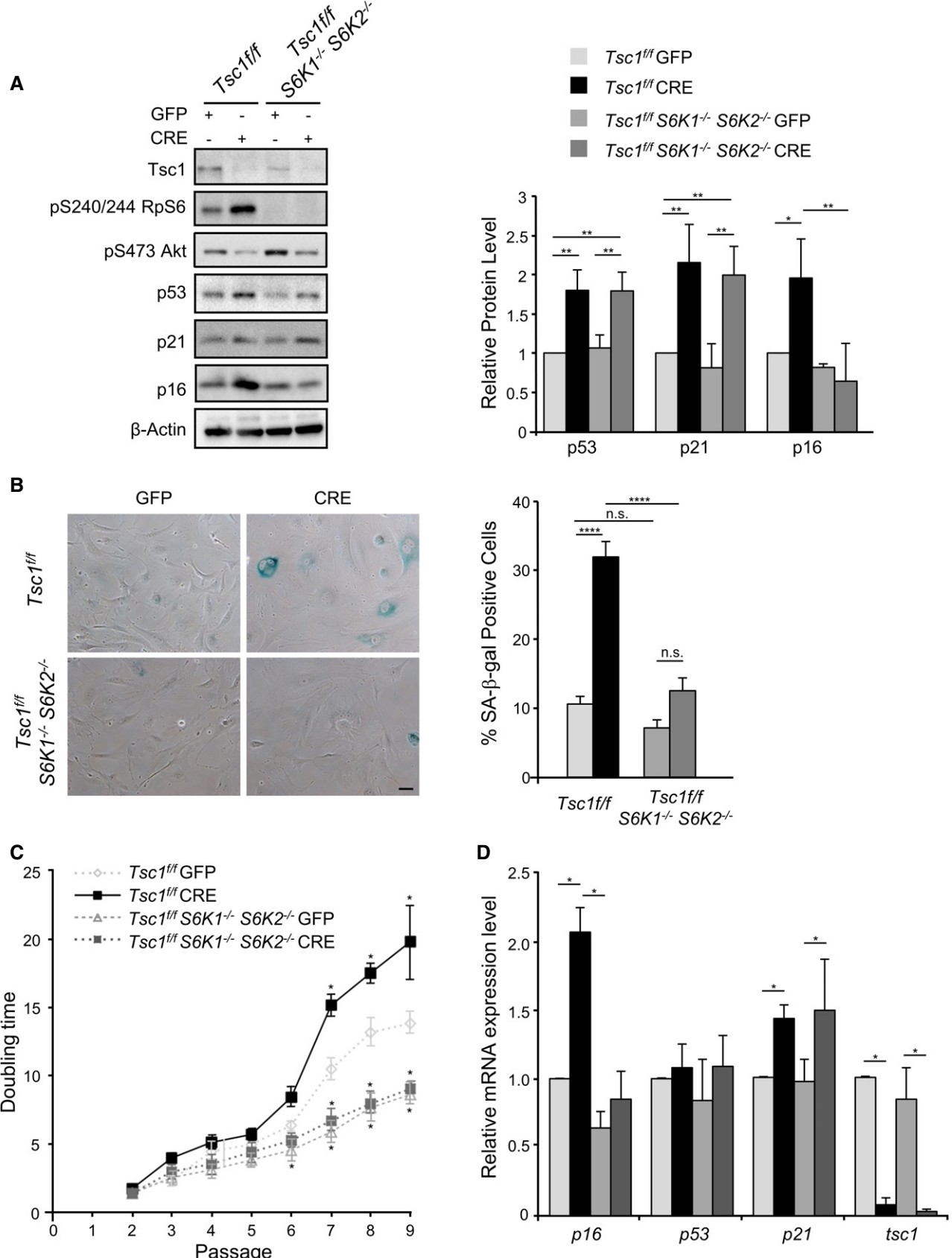

Figure 1.

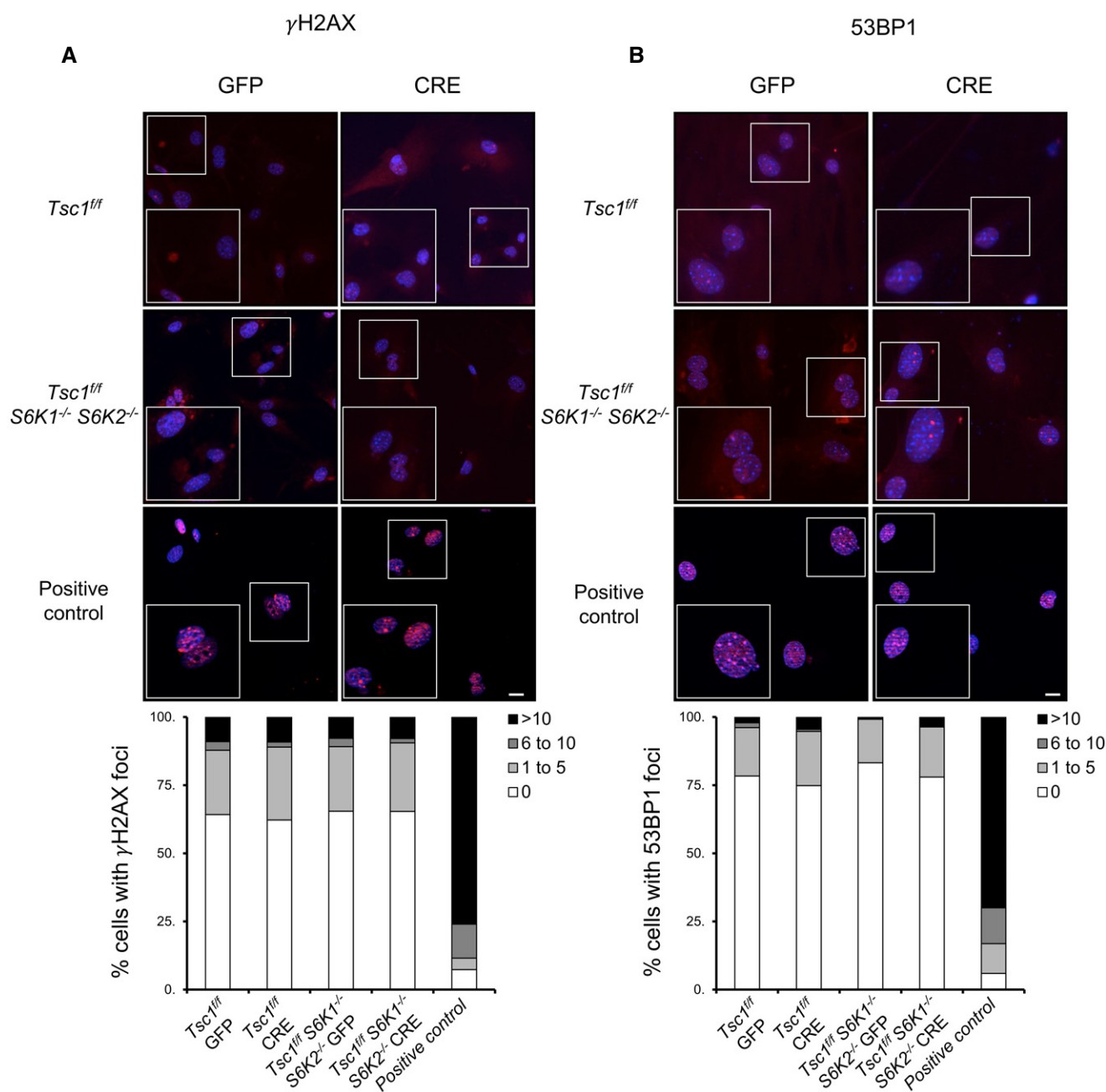

**Figure 2. TSC1 and S6 kinase deletion does not cause DNA damage-associated foci.**

A, B    $Tsc1^{fl/fl}$ and $Tsc1^{fl/fl}$ $S6K1^{-/-}S6K2^{-/-}$ primary MEFs were transduced with GFP or GFP-Cre adenovirus. Twelve days post-infection (passage P7) cells were fixed and analysed by immunofluorescence using the indicated antibodies. The formation of γH2AX or 53BP1 foci per cell was analysed in three independent experiments. Control cells treated with etoposide (6 μg/ml) overnight were used as a positive control ($n = 3$ cultures). Scale bars: 5 μm.

As an additional strategy to inhibit S6Ks, MEFs were treated with the selective S6K inhibitor PF-4708671 for 6 days starting at passage P5. The pharmacological treatment was sufficient to reduce senescence-associated β-galactosidase staining (Fig 3A) and p16 expression of TSC1 deleted cells both at protein and at RNA level (Fig 3B and C). The downregulation of p16 expression by lentiviral transduction of specific short hairpin RNA (shRNA) mimicked the effects of S6K inhibition on senescence-associated β-galactosidase staining

(Fig 3A). Moreover, the adenoviral-mediated overexpression of p16 restored the senescence programme in PF-4708671-treated TSC1 deleted cells (Fig 3A). Taken together, these data support the positive relationship between S6K activity and p16 expression in the control of TSC1 mutant cell senescence.

To find novel S6 kinase substrates potentially implicated in senescence, we set up a chemical genetic screen. The mutation of the ATP-binding pocket in protein kinases may open the access of

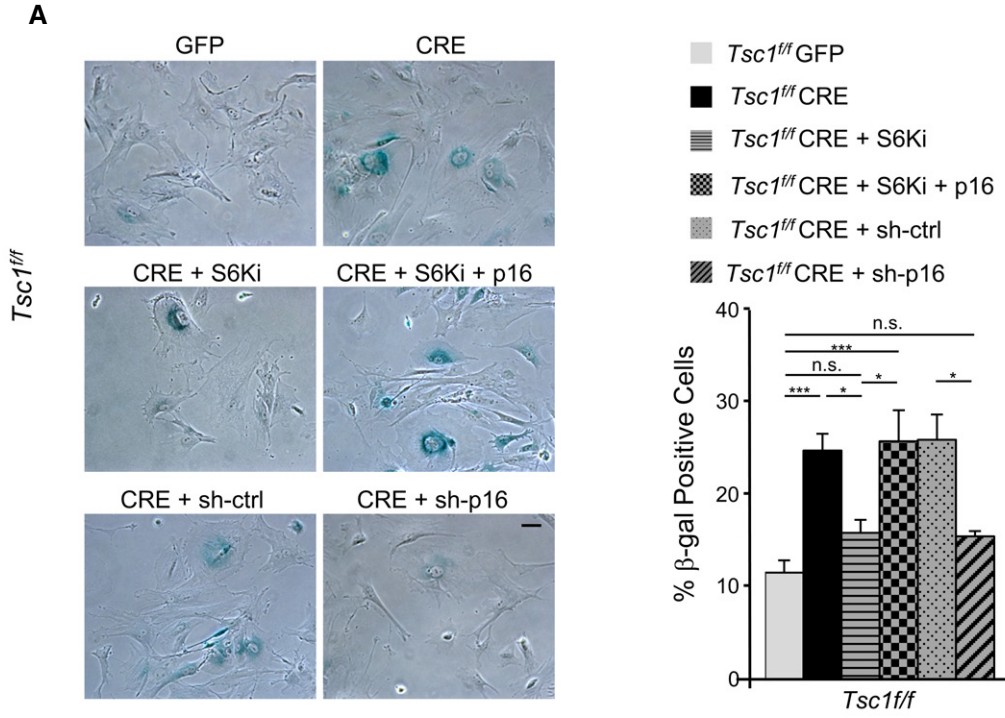

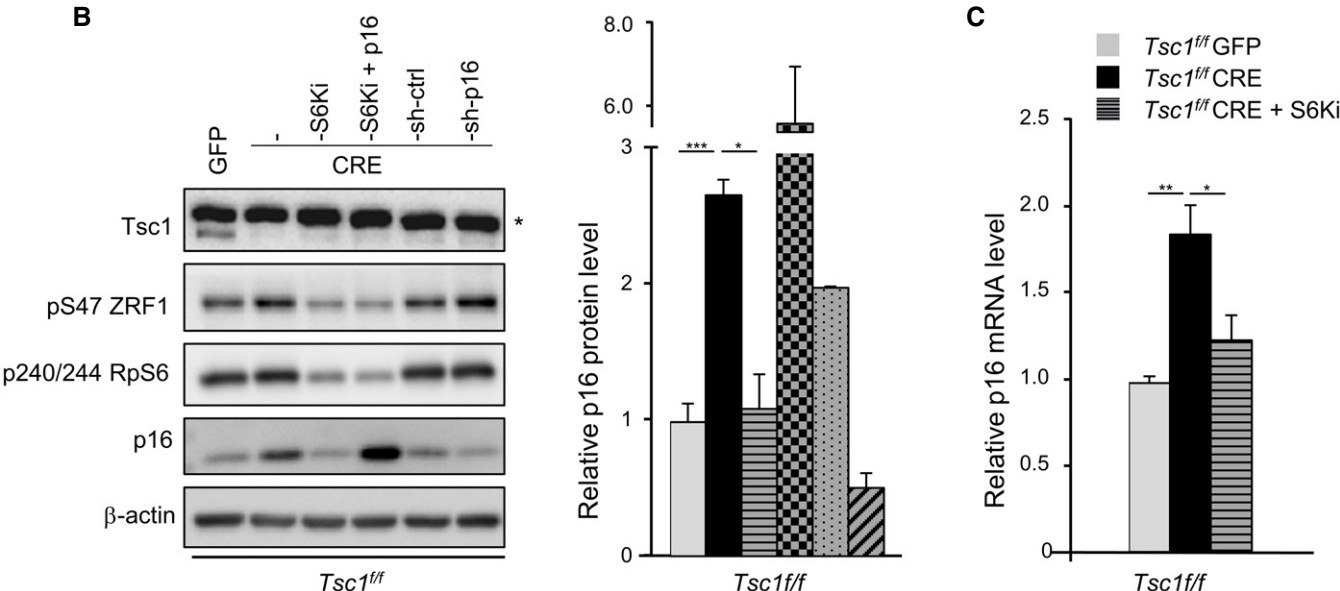

**Figure 3. p16 expression modulates the senescence programme during TSC1 deletion or S6K inhibition.**

A   Representative bright-field images of SA β-galactosidase staining performed on cells fixed at passage P7. After Cre-mediated TSC1 deletion, primary MEFs at passage P5 were treated with the S6K1 inhibitor PF-4708671 (10 μM), treated with PF-4708671 and simultaneously transduced with Adeno-GFP-p16 or transduced with LV-sh-p16 or LV-sh-Scramble, as control. PF-4708671 or DMSO was added every 24 h. Scale bar: 50 μm. Quantification of β-galactosidase-positive cells was performed on three independent experiments and is shown in the graph on the right. Data are presented as mean ± SEM; *P < 0.05, ***P < 0.001, ANOVA multiple comparisons.

B   Representative immunoblot analyses of MEF primary cells treated as in (A) and harvested at passage P7. * indicates unspecific band. Densitometric analyses of actin-normalized signal are presented in a graph as fold changes relative to the control. Data are presented as mean ± SEM of three experiments. *P < 0.05, ***P < 0.001, two-tailed, unpaired Student's t-test.

C   Relative transcript level of p16 of MEF primary cells treated as in (A) and harvested at passage P7. Data are presented as mean ± SEM of three experiments. *P < 0.05, **P < 0.01, ANOVA multiple comparisons.

the catalytic domain to ATP thiophosphate analogues harbouring substitutions with bulky residues, which cannot be used by the majority of wild-type kinases. After the kinase reaction with ATP thiophosphate analogues in digitonin-permeabilized cells, the alkylation of thiophosphorylated proteins and the detection with a thiophosphate-ester-specific antibody allows us to assess whether a protein is a direct substrate of the mutant kinase in a cellular environment (Hertz *et al*, 2010; Banko *et al*, 2011). To adapt the technique for S6K, we mutated the gatekeeper residue Lys149 of S6K1 to Gly and compared the ability of the mutant kinase vs. wild type to thiophosphorylate the well-characterized S6K substrate ribosomal protein S6 (rpS6) by using benzyl-ATP thiophosphate in human embryonic kidney HEK293 cells. As shown in Fig 4A, the mutant

kinase, but not the endogenous nor the ectopically expressed wild-type kinase (WT-S6K1), was able to use the ATP analogue and thiophosphorylate HA-tag rpS6. The mutation of the Ser residues to Ala in the carboxy-terminal tail of HA-tag rpS6 was sufficient to abolish thiophosphorylation, suggesting that the mutant kinase is active and specifically recognize the S6K phosphorylation sites. The mutant kinase was therefore named analogue-specific S6K1 (AS-S6K1).

A series of S6K substrates, including rpS6, CAD, eIF4B, PDCD4, eEF2K, SKAR, IRS1, were tested for thiophosphorylation by AS-S6K1 and WT-S6K1 in HEK293 cells (Fig 4B). Generally, the expression of AS-S6K1 resulted in a sharp increase in detection by the thiophosphate-ester-specific antibody after *in situ* kinase reaction and derivatization, as compared to WT-S6K1. For a minority of

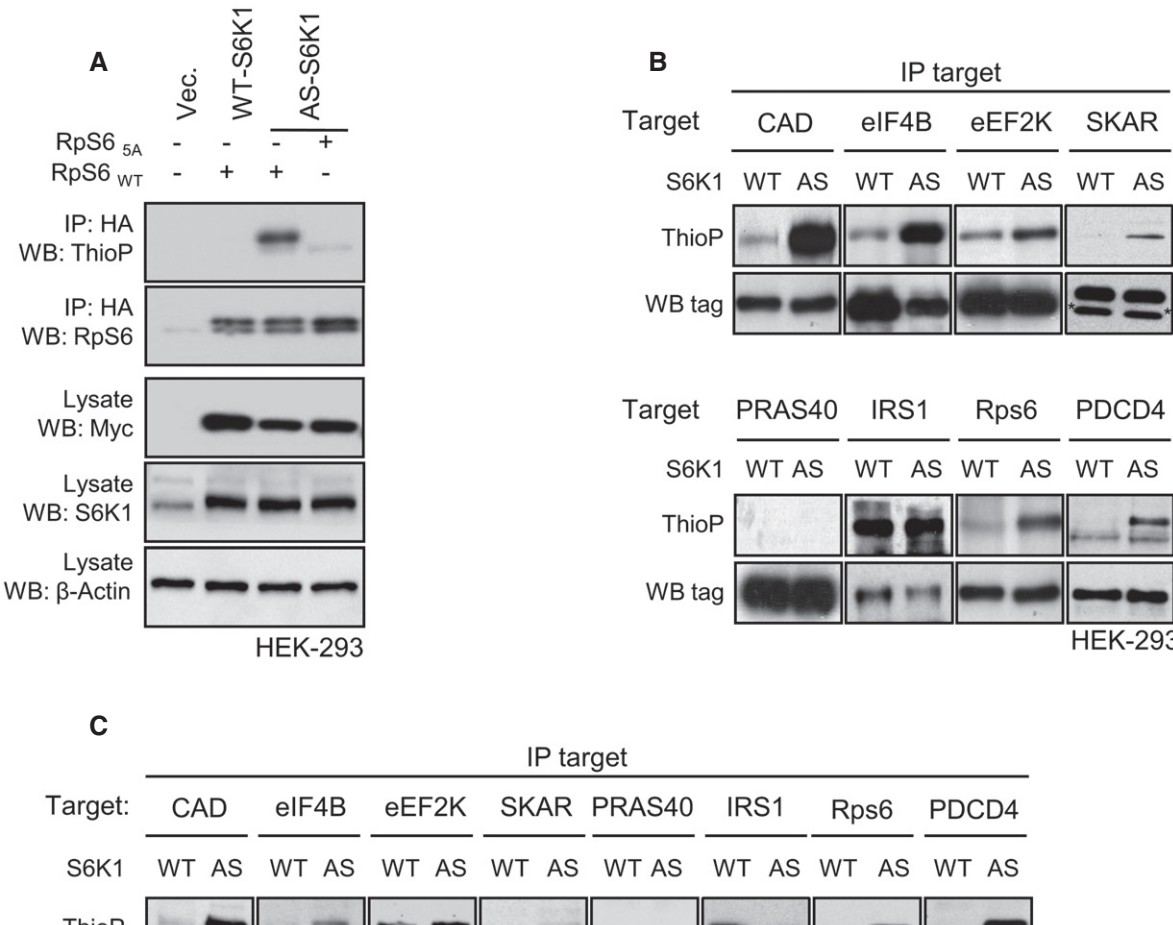

**Figure 4.  Validation of analogue-specific S6K1 mutation.**

A    Myc-WT-S6K1 and myc-AS-S6K1 were transfected in HEK293 cells with HA-RpS6WT or HA-RpS65A. *In vivo* kinase assay was performed in the presence of 6-Bn-ATP-γ-S. After immunoprecipitation using an anti-HA antibody, the thiophosphorylation of RpS6 was revealed by Western blot using an anti-thiophosphate ester antibody. Expression level of myc-WT-S6K1 and myc-AS-S6K1 or total S6K1 was revealed by Western blot on total extracts using the indicated antibody.

B, C    HEK293 or U2OS cells stably expressing myc-WT-S6K1 (WT) or myc-AS-S6K1 (AS) were transfected with tagged forms of S6K1 substrates (CAD, eIF4B, eEF2K, SKAR, IRS1, RpS6, PDCD4) or PRAS40. *In vivo* kinase assay was performed in the presence of 6-Bn-ATP-γ-S. After immunoprecipitation, the thiophosphorylation was revealed by Western blot using an anti-thiophosphate ester antibody. * indicates unspecific band.

targets, such as IRS1 and eEF2K, a relatively high background signal was already detected in WT-S6K1 expressing cells, which in the case of IRS1 was not further increased in AS-S6K1 expressing cells. Despite these exceptions, the AS-S6K1 can promote thiophosphorylation for the large majority of reported S6K substrates. Importantly, PRAS40 protein, which contains multiple residues phosphorylated by mTOR and Akt but not S6K (Sancak *et al*, 2007), was not thiophosphorylated by AS-S6K1. This was particularly encouraging as Akt and S6K share a similar consensus site for phosphorylation, with Ser-Thr residues surrounded by basic residues in the −3 and −5 positions. These results were confirmed in another cell type, *that is* the human osteosarcoma U2OS line (Fig 4C). Therefore, although some background noise may preclude the purification of endogenous substrates in the whole proteome, the AS-S6K1 is a valuable tool to reveal direct phosphorylation of candidate proteins in the cellular environment.

Quantitative proteomic analyses have recently identified hundreds of proteins that are differentially phosphorylated after pharmacological treatment with rapamycin or mTOR catalytic inhibitors, or after genetic manipulation of TORC1/TORC2 components (Moritz *et al*, 2010; Hsu *et al*, 2011; Yu *et al*, 2011; Robitaille *et al*,

2013). From these databases, we selected a list of 10 proteins harbouring a putative S6K consensus site for phosphorylation and a reported function in cell cycle control, senescence and insulin sensitivity. This short list included zuotin-related factor 1 (ZRF1), homeobox protein cut-like 1 (Cux1), N-myc downstream-regulated gene 1 protein (NDRG1), nucleophosmin (NPM1), La-related protein 1 (Larp1), lamin A, JunB, 75-kDa glucose-regulated protein (Grp75), euchromatic histone-lysine *N*-methyltransferase 2 (EHMT2) and enhancer of mRNA-decapping protein 3 (EDC3). Next, we asked whether a Flag-tagged version could be directly thiophosphorylated by AS-S6K1. As an arbitrary cut-off to select for *bona fide* substrates, the following criteria were used (i) protein thiophosphorylation is increased more than fivefold in AS-S6K1 expressing cells as compared to WT-S6K1 expressing cells, and (ii) protein thiophosphorylation is confirmed in two distinct cell types. NPM1 (Fig 5A), NDRG1, Larp1, JunB, Grp75, EHMT2 and EDC3 (data not shown) gave negative results, while ZRF1, Cux1 and lamin A matched these criteria (Fig 5A) and were therefore selected as bona fide novel substrates of S6K1.

The known biological function of ZRF1 (aka DNAJC2, MIDA1 and Mpp11) drew our attention as a particularly interesting target,

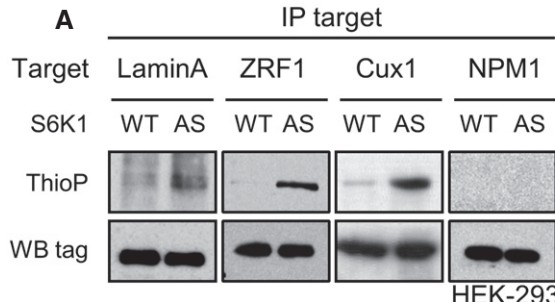

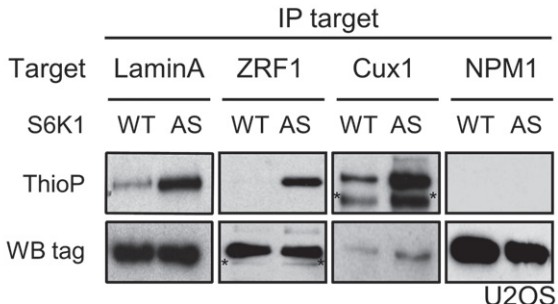

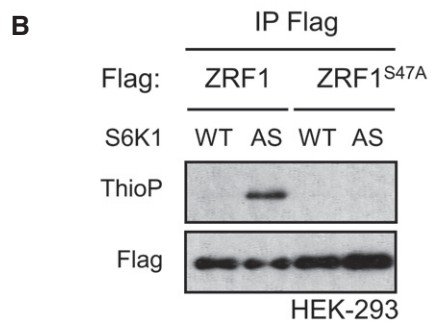

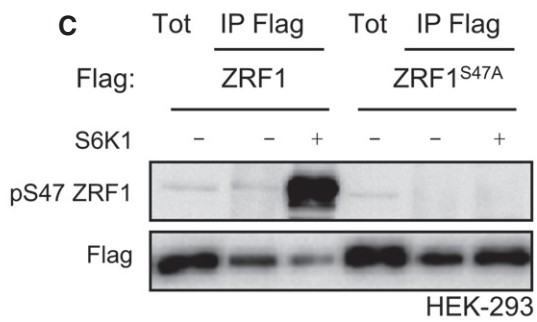

**Figure 5. Analogue-specific S6K1 mutation to screen for direct *in vivo* substrates.**

A HEK293 cells stably expressing myc-WT-S6K1 (WT) or myc-AS-S6K1 (AS), or U2OS transduced with adeno-myc-WT-S6K1 (WT) or adeno-myc-AS-S6K1 (AS), were transfected with tagged forms of candidates (lamin A, ZRF1, CUX1, NPM1). *In vivo* kinase assay were performed in the presence of 6-Bn-ATP-γ-S. After immunoprecipitation, the thiophosphorylation was revealed by Western blot using an anti-thiophosphate ester antibody. * indicates unspecific band.

B HEK293 cells stably expressing myc-WT-S6K1 (WT) or myc-AS-S6K1 (AS) were transfected with Flag-tagged forms of ZRF1 or a mutant of ZRF1, ZRF1^S47A. *In vivo* kinase assay was performed in the presence of 6-Bn-ATP-γ-S. After immunoprecipitation using an anti-Flag antibody, the thiophosphorylation was revealed by Western blot using an anti-thiophosphate ester antibody.

C HEK293 was transfected with Flag-ZRF1WT or Flag-ZRF1S47A mutant plasmids. Twenty-four hours post-transfection, cells were starved overnight and treated for 3 h with Torin 1 (100 nM). After immunoprecipitation with anti-Flag antibody, an *in vitro* kinase assay was performed with a recombinant active S6K1. ZRF1 phosphorylation was analysed by immunoblotting.

**Figure 6.  ZRF1 is a novel S6 kinase substrate in mouse tissues *in vivo* and in primary MEFs.**

A   Immunoblot analysis of proteins extracted from liver of WT mice that were starved overnight and refed for 4 h. When indicated, mice were injected intraperitoneally with 5 mg/kg of the rapamycin derivate temsirolimus 1 h before refeeding (*n* = 3 mice for each condition).

B   Immunoblot analysis of proteins extracted from liver and WAT of WT and *S6K1*$^{-/-}$*S6K2*$^{-/-}$ mice that were starved overnight or starved overnight and refed for 4 h. Proteins were analysed by Western blot using the indicated antibodies (*n* = 3 mice for each genotype).

C   Immunoblot analysis of proteins extracted from liver of WT, *S6K1*$^{-/-}$, *S6K2*$^{-/-}$ and *S6K1*$^{-/-}$*S6K2*$^{-/-}$ mice starved overnight and refed for 4 h (*n* = 3 mice for each genotype).

D   Representative immunoblot of *Tsc2*$^{-/-}$*p53*$^{-/-}$ and *Tsc2*$^{+/+}$*p53*$^{-/-}$ immortalized MEFs kept in serum, starved for serum overnight or starved for serum overnight and amino acids for 2 h. Proteins extracts were analysed by Western blot using the indicated antibodies (*n* = 2 cultures). Densitometric analyses of phosphorylation level of ZRF1 normalized to total tubulin are shown in Fig EV2B.

E   Representative immunoblot analysis of ZRF1 phosphorylation levels in *Tsc1*$^{fl/fl}$ and *Tsc1*$^{fl/fl}$ *S6K1*$^{-/-}$*S6K2*$^{-/-}$ primary MEFs transduced with GFP or GFP-Cre adenovirus and harvested at passage P7. Densitometric analyses of phosphorylation level of ZRF1 normalized to total β-actin protein levels are shown in Fig EV2C.

as in the nucleus ZRF1 may act as an epigenetic factor or transcriptional co-regulator influencing cell fate determination and senescence (Richly *et al*, 2010; Ribeiro *et al*, 2013; Aloia *et al*, 2014). Cytosolic ZRF1 may act as a co-chaperon forming the ribosome-associated complex (RAC) together with heat-shock 70-kDa protein 14 (HspA14) for the proper folding of nascent polypeptides at the tunnel exit of ribosomes (Hundley *et al*, 2005; Jaiswal *et al*, 2011). We therefore decided to further validate whether ZRF1 is a novel substrate of S6K. First, we determined the phosphorylation site in the ZRF1 protein by AS-S6K1. Human ZRF1 contains a Ser47 that harbours a RXRXXS motif and has been previously shown to be differentially phosphorylated following wortmannin, rapamycin and Torin treatments (Moritz *et al*, 2010; Hsu *et al*, 2011; Robitaille *et al*, 2013). This residue is conserved across vertebrates (Fig EV1). Interestingly, unicellular organisms, such as yeast, have a threonine residue in the homologous region that has been shown to be differentially phosphorylated after rapamycin treatment (Claudio de Virgilio, personal communication). Mutation of Ser47 to Ala in the mouse protein sequence was sufficient to completely abrogate thiophosphorylation by AS-S6K1 (Fig 5B), indicating that this is likely to be the unique phosphorylation site. Next, we addressed whether ZRF1 was a direct substrate of recombinant S6K in *in vitro* kinase reactions. Unphosphorylated Flag-tag wild-type ZRF1 or the Ser47 to Ala mutant was immunopurified from protein extracts of HEK293 cells that were previously transfected with ZRF1-coding plasmids and treated with the mTOR inhibitor Torin 1. As shown in Fig 5C, recombinant S6K1 protein was able to phosphorylate wild-type ZRF1, but not the Ser47 to Ala mutant, as detected by using a ZRF1 Ser47 phospho-specific antibody.

Next, endogenous ZRF1 phosphorylation by endogenous S6K was analysed *in vivo* and in cultured cells. In liver tissue from wild-type mice, ZRF1 Ser47 phosphorylation was sensitive to mTOR inhibition by rapamycin treatment (Fig 6A). In addition, overnight starvation and 3 h-refeeding, respectively, down- and upregulated ZRF1 phosphorylation in liver, white adipose tissue (WAT) and skeletal muscle (Figs 6B and EV2A). Strikingly, the combined deletion of S6 kinases 1 and 2 abolished ZRF1 phosphorylation, similar to rpS6 phosphorylation (Figs 6B and EV2A). However, S6K1 and S6K2 appeared to have relatively distinct activities towards rpS6 and ZRF1. As expected, rpS6 was a preferential substrate for S6K2, while S6K1 deletion had more potent effects than S6K2 on phospho-ZRF1 vs. phospho-rpS6 (Fig 6C). In TSC2-deficient immortalized cells, ZRF1 and rpS6 phosphorylation was insensitive to serum and amino acid withdrawal, as opposed to wild-type *Tsc2*$^{+/+}$ cells (Figs 6D

and EV2B). Importantly, in the same experimental setting to study premature senescence, *that is* primary MEFs after adenoviral Cre-mediated deletion of the TSC1-floxed allele (Fig 1A), the increased ZRF1 phosphorylation was abrogated by the concomitant deletion of S6K1/2 (Figs 6E and EV2C) and by the S6K inhibitor PF-4708671 (Fig 3B). Taken together, our data delineate a pathway leading to ZRF1 phosphorylation by S6K1 and S6K2 in response to nutrient availability or as a consequence of genetic perturbations activating mTORC1.

The rapamycin-sensitive phosphorylation of ZRF1 was detected in both cytosolic and nuclear fractions (Fig 7A). Rapamycin treatment did not affect nuclear translocation nor protein levels. A putative function for nuclear ZRF1 is the regulation of oncogene-induced senescence (Ribeiro *et al*, 2013). To evaluate the role of ZRF1 phosphorylation, endogenous murine ZRF1 was depleted by lentiviral shRNA transduction followed by rescue with human wild-type or phospho-mutant ZRF1 (Fig 7B and C). In the functional studies with the phospho-mutant, both Ser47 and Ser49 were mutated to Ala, to avoid the possibility of a residual phosphorylation in the nearby Ser residue. As previously shown in other cell types and conditions (Ribeiro *et al*, 2013), ZRF1 knock-down (KD) decreased senescence-associated β-galactosidase staining in both TSC1-floxed control and deficient cells (Fig 7B). Overexpression of wild-type ZRF1 rescued the senescence programme while the phospho-mutant was unable to trigger this response. Importantly, the senescence programme dependent on ZRF1 expression and phosphorylation correlated with p16 levels, which were low in ZRF1 KD or phospho-mutant cells (Fig 7C).

## Discussion

Our study provides a method to rapidly screen putative S6K substrates by direct *in vivo* labelling. Quantitative phospho-proteomic studies have recently led to the deep and high-quality coverage of phosphopeptides whose abundance varies with perturbations in mTOR signalling (Moritz *et al*, 2010; Hsu *et al*, 2011; Yu *et al*, 2011; Robitaille *et al*, 2013). Strikingly, the phosphorylation of approximately 4% of human gene products is under control of the mTOR pathway. The phosphopeptides include direct mTOR substrates but also proteins whose phosphorylation is controlled by kinases and phosphatases downstream of mTOR. This information, in combination with *in silico* searches for Ser/Thr sites surrounded by basic residues, provides a powerful tool to interrogate the mTOR phospho-proteome and reveals a subset of S6K direct substrates

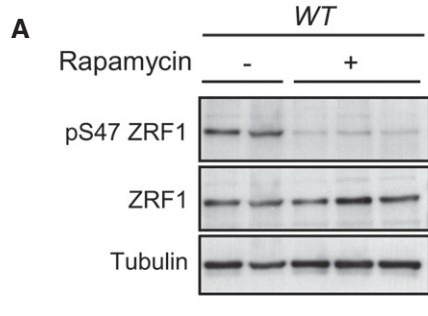

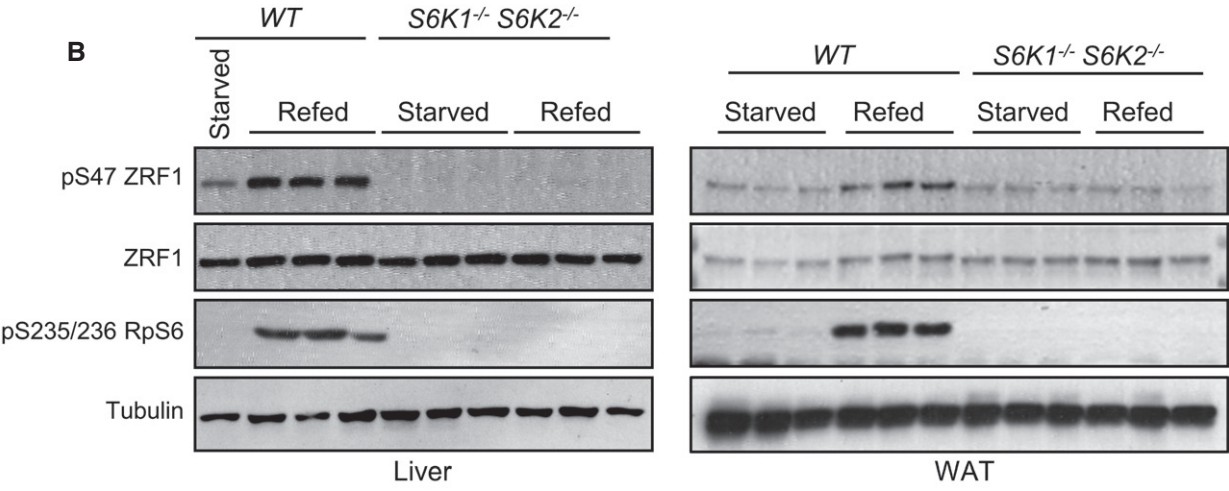

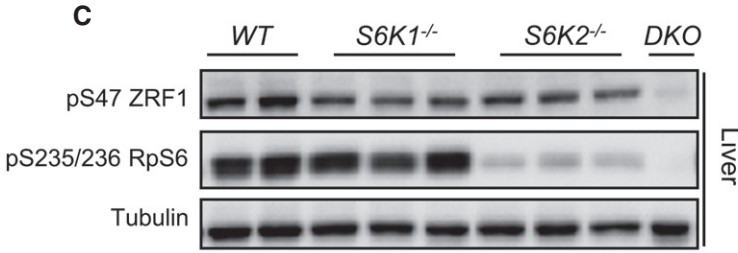

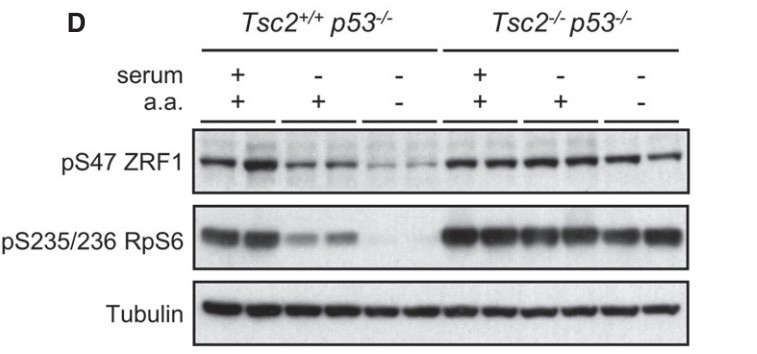

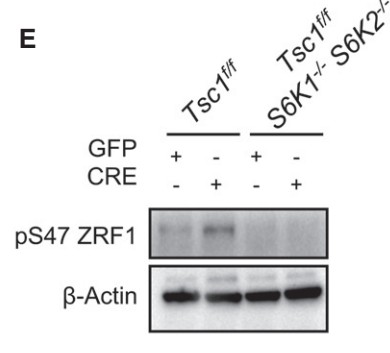

**Figure 6.**

using an analogue-specific mutant kinase. In this initial study, out of ten putative candidates, three proteins are reliably phosphorylated by AS-S6K1: Cux1 is a transcription factor involved in tumour suppression and the retrograde control of insulin signalling (Wong *et al*, 2014); lamin A is a component of the nuclear envelope that may affect ageing (Ghosh & Zhou, 2014); and ZRF1, whose

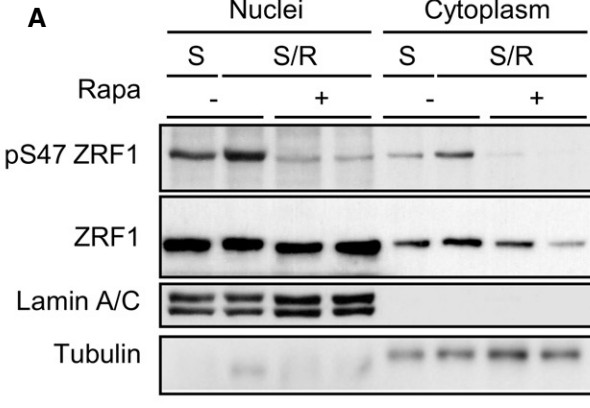

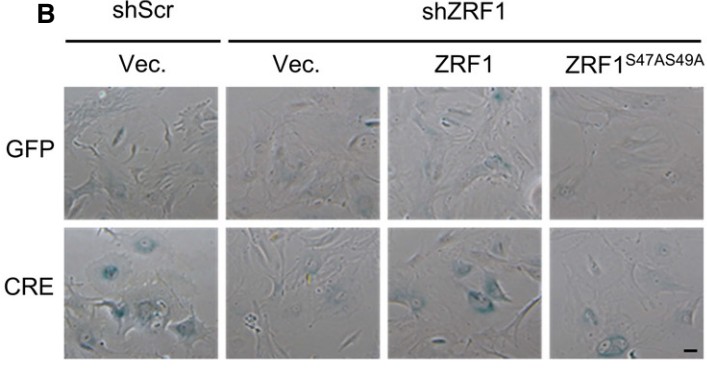

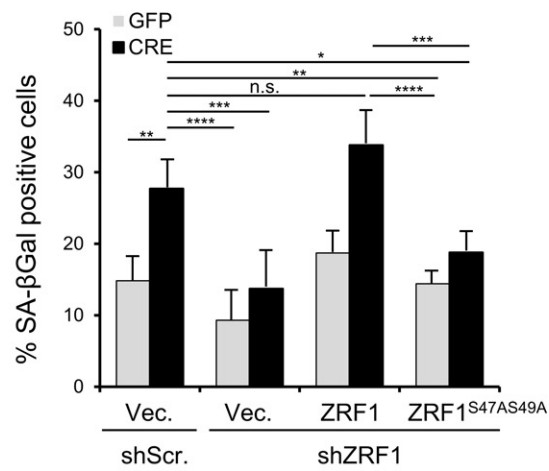

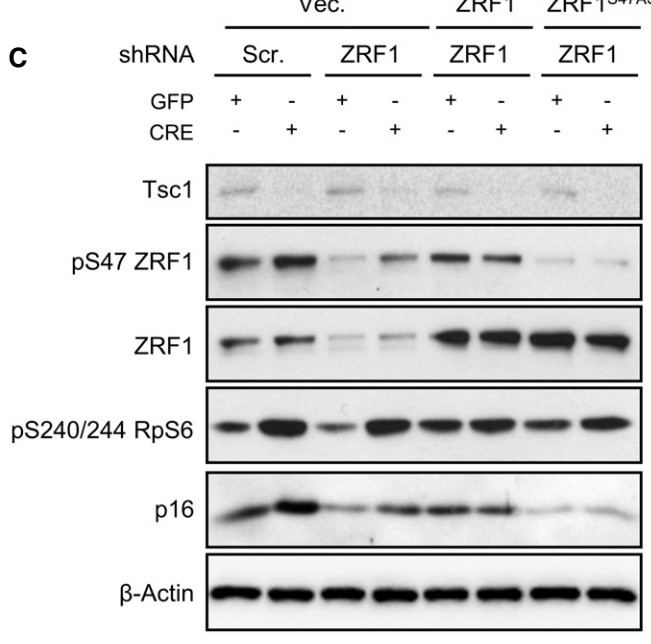

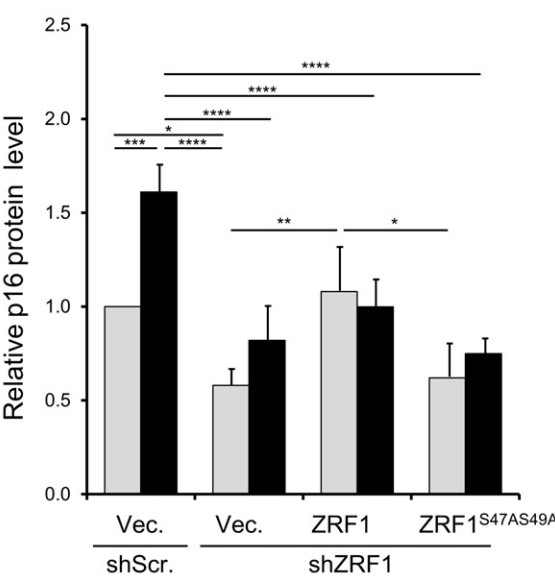

**Figure 7.**

phosphorylation has been more thoroughly characterized here. After validation with phospho-specific antibodies recognizing the endogenous protein, we demonstrate that ZRF1 is a novel substrate of

S6Ks. The phosphorylation of ZRF1-Ser47 by S6Ks participates in a senescence programme through the regulation of the cell cycle inhibitor p16.

**Figure 7. The ZRF1 phosphorylation mutant blunts the senescence programme.**

A   Immunoblot analysis of nuclear and cytoplasmic fractions extracted from liver of WT mice starved overnight and refed for 4 h. When indicated, mice were injected intraperitoneally with 5 mg/kg of the rapamycin derivative temsirolimus 1 h before refeeding.

B   Representative bright-field images of SA β-galactosidase staining. $Tsc1^{fl/fl}$ primary MEFs (passage P1) were transduced with lentivirus shScramble or shZRF1 followed by transduction with lentivirus expressing ZRF1 or ZRF1 A mutant ($ZRF1^{S47AS49A}$) or the empty vector. Cells were finally transduced with GFP or GFP-Cre adenovirus (at passage P3). Seven to nine days post-adenoinfection (P6 and P7), cells were fixed and β-galactosidase staining was performed. Quantification of β-galactosidase-positive cells was performed on three independent experiments. Data are presented as mean ± SEM; *$P < 0.05$, **$P < 0.01$, ***$P < 0.001$, ****$P < 0.0001$, n.s. not significant, two-way ANOVA, Tukey's multiple comparisons test.

C   $Tsc1^{fl/fl}$ primary MEFs were transduced with lentivirus shScramble or shZRF1 followed by transduction with lentivirus expressing ZRF1 or ZRF1 A mutant ($ZRF1^{S47AS49A}$) or the empty vector. Cells were finally transduced with GFP or GFP-Cre adenovirus and proteins were analysed by Western blot using the indicated antibodies. Data are expressed as relative values normalized to β-actin, with respect to the control. Data are reported as mean ± SEM of $n = 3$; *$P < 0.05$, **$P < 0.01$, ***$P < 0.001$, ****$P < 0.0001$, two-way ANOVA, Tukey's multiple comparisons test.

Loss-of-function mutations in TSC1/TSC2, which lead to unrestrained mTOR activity, induce premature senescence (Zhang et al, 2003). mTOR is required for the senescence response as the allosteric inhibitor rapamycin blunts this process. It has been proposed that rapamycin promotes quiescence, a reversible cell cycle arrest as opposed to the irreversible arrest during senescence. The mTOR substrates 4EBP1 and 4EBP2 have been previously shown to be involved in senescence, downstream of mTORC1 (Petroulakis et al, 2009). 4EBPs are negative regulators of cap-dependent translation through their interaction with the translation initiation factor eIF4E. mTOR phosphorylation of 4EBPs relieves this inhibition, thus allowing the binding of eIF4E to eIF4G. 4EBP1/4EBP2-deficient cells undergo faster replicative senescence as compared to wild-type control. Interestingly, the 4EBP branch of mTORC1 signalling regulates the p53-dependent senescence response through the translation control of the p53-stabilizing protein Gas2, Arf or p53 itself (Petroulakis et al, 2009; Alimonti et al, 2010). Conversely, 4EBP1/4EBP2-deficient cells do not display increased p16 levels (Petroulakis et al, 2009). S6Ks also participate in the cell cycle arrest and a senescence response as inferred by shRNA approaches or pharmacological inhibition (Nicke et al, 2005; Leontieva et al, 2013). However, molecular and functional mechanisms were not addressed. Here, we show reduced senescence in S6K-deficient cells after TSC1 deletion. Of note, this is accompanied by impaired expression of the cell cycle inhibitor p16, while the p53/p21 response is unaffected. Therefore, mTORC1 signalling coordinates the senescence programme through the complementary action of 4EBPs and S6Ks that selectively control p53/p21 and p16, respectively.

The ZRF1 protein has previously been shown to regulate differentiation and senescence programmes in the nucleus (Richly et al, 2010; Ribeiro et al, 2013; Aloia et al, 2014). ZRF1 contains a SANT domain for DNA and histone binding, as well as an ubiquitin binding domain (UBD). It has been proposed that ZRF1 binds monoubiquitinated histone H2A on Lys119 and favours its deubiquitination (Richly et al, 2010). ZRF1 can therefore antagonize the action of polycomb repressive complex 1 (PRC1) that ubiquitinates H2A through the E3 ubiquitin ligase activity of Ring1B and silences genes involved in senescence and differentiation, such as the Ink4a/ARF locus encoding p16. By west-Western screening, ZRF1 has also been shown to interact with inhibitor of differentiation 1 (Id1) and, for this reason, is also known as Mida1 (mouse Id1-associated protein) (Shoji et al, 1995). Depending on the cell type and environmental conditions, Id1 can inhibit quiescence, differentiation and senescence. Id family members are helix-loop-helix (HLH) proteins that lack a DNA binding domain but can heterodimerize with basic HLH (bHLH) transcription factors and suppress their activity (Lasorella et al, 2014). Different bHLH proteins are master regulators of cell differentiation along a variety of lineages. Id proteins can also suppress the activity of ETS1 and ETS2 transcription factors, which are potent activators of p16 transcription. Hence, Id1 and p16 levels usually display a negative correlation, and Id1-deficient MEFs have higher p16 levels and senescence rates. Thus, ZRF1 may control p16 expression through at least two distinct and converging mechanisms, the regulation of PRC1 and Id1. From analysis of the ZRF1 phospho-mutant, we demonstrate that ZRF1 phosphorylation on Ser47 participates in signal transduction from mTORC1/S6Ks to p16 and the senescence programme. It will be interesting to address whether this is through PRC1 activity, Id1 regulation and/or other alternative mechanisms. For instance, at this stage, we also cannot exclude a contribution of cytosolic ZRF1.

It is likely that ZRF1 is not the sole S6K1/2 target involved in the senescence programme. Mdm2 can be phosphorylated by S6K and Akt in conditions of doxorubicin-induced DNA damage that activate a p38 stress response (Lai et al, 2010). Another substrate, lamin A, which we uncovered by AS-S6K1 screening, could also play a role in ageing, although this level of regulation awaits functional validation (Ghosh & Zhou, 2014). Although a certain degree of redundancy is predictable, our current knowledge also indicates striking functional specialization in the branches downstream of mTORC1, as shown here for S6Ks selectively controlling the p16 programme. Intriguingly, p16 levels are upregulated in multiple mouse and human tissues during ageing and inversely correlate with a capacity for tissue regeneration (Sousa-Victor et al, 2014). S6K activity also increases during ageing and has been negatively associated to lifespan. In future, this wealth of information could help implement strategies to affect tissue senescence, quiescence, regeneration and reprogramming in diseases, including cancer- and age-related syndromes.

# Materials and Methods

### Reagents

The constructs encoding Flag-CAD, Flag-eIF4B, Flag-ZRF1 were generated by subcloning the respective cDNAs obtained from the Orfeome library of cDNA (Open Biosystems/Thermo Scientific, CAD: MHS1010-9206052; eIF4B: MHS1010-98051322; ZRF1: MMM1013-63294) into the pcDNA3.1 mammalian expression vector in frame with the Flag-epitope tag. The constructs encoding

Flag-eEF2K, Flag-IRS1, Flag-PDCD4, Flag-lamin A were generated by subcloning the respective cDNAs obtained from Addgene (https://www.addgene.org, ADDGENE 23726, ADDGENE 11025, ADDGENE 20693, ADDGENE 17662) into the pcDNA3.1 mammalian expression vector in frame with the Flag-epitope tag. The PADDOX-RpS6 vector, pcDNA3-Flag-SKAR and pCMV-Tag2A-Cux1 were kindly provided by Stefano Fumagalli (INSERM 1151, France), John Blenis (Weill Cornell Medical College, USA) and Jaquelin P. Dudley (University of Texas, USA), respectively. The human ZRF1 plasmid pCMV2A-ZRF1 was kindly provided by Luciano Di Croce (CRG, Spain) and cloned into the pcDNA3.1 mammalian expression vector in frame with the Flag-epitope tag. pRIPU-S6K1 L149G (AS-S6K1), pcDNA3-Flag-ZRF1 S47A and pcDNA3-Flag-ZRF1 S47AS49A mutants were generated by site-directed mutagenesis using the QuikChange site-directed mutagenesis kit (Stratagene). The presence of the specific mutations was verified by sequencing. Adenoviral vector expressing WT-S6K1 and AS-S6K1 were generated and amplified by VectorBiolabs (USA). GFP and GFP-Cre adenoviral vectors were described previously (Nemazanyy *et al*, 2015). Lentiviral plasmids pLKO.1-shZRF1 (Sigma-Aldrich, TRCN0000374114) and pLKO.1-shRNA scrambled (Sigma-Aldrich, SHC016) were produced by the VVTG platform (SFR Necker, France). Lentiviral plasmids encoding ZRF1 and ZRF1$^{S47AS49A}$ resistant to murine shRNA were generated by subcloning the respective human cDNAs into the pLEX-MCS vector and were produced by the VVTG platform (SFR Necker, France). Adenoviral vector expressing murine p16 was purchased from VectorBiolabs (USA). p16 sh-RNA lentiviral plasmid was a kind gift from Pura Muñoz-Cánoves (UPF, Spain). The following primary antibodies were used: tubulin (Sigma, T9026); β-actin (Sigma, A5316); pSer473 Akt (Cell Signaling Technology, 9275); lamin A/C (Cell Signaling Technology, 2032); Flag-tag M2 (Sigma, F3162); thiophosphate ester antibody (Abcam, ab92570); HA-epitope tag (Pierce, 26183); DNAJC2/MPP11 (Cell Signaling Technology, 12844); pSer47 DNAJC2/MPP11 (Cell Signaling Technology, 12397); Myc-tag (Cell Signaling Technology, 2276); S6 ribosomal protein (Cell Signaling Technology, 2217); pSer235/236 S6 ribosomal protein (Cell Signaling Technology, 2211), pSer240/244 S6 ribosomal protein (Cell Signaling Technology, 5364); TSC1 (Cell Signaling Technology, 4906); p53 (Novocastra Laboratories, NCL-p53-CM5p); p21 (Cell Signaling Technology, 2946), p16 (Santa Cruz Biotechnology, sc-1267), 53BP1 (Santa Cruz Biotechnology, sc-22760), γH2AX (Bethyl laboratories, A300-081A).

PF-4708671 and Torin were obtained from Sigma-Aldrich and Tocris, respectively. Stock solutions were dissolved in DMSO.

## Cell culture

Mouse embryonic fibroblasts were prepared from embryos at embryonic day 13.5 as previously described (Shima *et al*, 1998). Briefly, embryos were minced and incubated in 0.25% trypsin at 37°C for 30 min and then passed through a cell strainer (Falcon). Dulbecco's modified Eagle's medium (DMEM), containing 10% foetal calf serum (FCS), was added to the cell suspension. Cells were then centrifuged at 195 *g*, and the pellet was suspended in DMEM containing 10% FCS. To obtain *Tsc1*$^{-/-}$ MEFs, fibroblasts were transduced at 150 MOI by adenoviruses GFP-Cre (GFP was used as a control) and incubated for the indicated time before analyses.

For silencing and re-expression experiments, primary MEFs were first transduced with a lentivirus expressing shZRF1 at a MOI of 10 with polybrene (8 μg/ml). After 24 h, the medium was changed and cells were selected for 48 h in 3 μg/ml puromycin. To overexpress ZRF1 WT or A mutant, cells were infected 4 days following the first infection with lentiviruses at 10 MOI with polybrene (8 μg/ml). After 24 h, cells were selected for 48 h in 3 μg/ml puromycin. The *Tsc1* gene was deleted by adenoviral infection. For the pharmacological inhibition of S6 kinase, starting from passage P5, Cre-mediated TSC1-deleted MEF cells were treated every 24 h with PF-4708671 (10 μM) or DMSO as control. For the rescue experiment with p16, TSC1-deleted cells at passage P5 were simultaneously treated with PF-4708671 and transduced with adenoviral particles expressing p16. To explore the effect on senescence of silencing p16 gene, TSC1-deleted primary MEFs were transduced at passage P5 with sh-RNA-p16 or with control shScramble lentiviral particles, at MOI 10 with polybrene (8 μg/ml). Cells at passage P7 were fixed for SA β-galactosidase staining or harvested for protein or RNA extraction.

*Tsc2*$^{+/+}$; *p53*$^{-/-}$ and *Tsc2*$^{-/-}$*p53*$^{-/-}$ immortalized MEFs (kind gift of D.J. Kwiatkowski, Brigham and Women's Hospital, Boston, MA, USA) as well as HEK293 cells and U2OS cells were cultured in DMEM with 10% FBS and 1% penicillin/streptomycin. All cells were cultured at 37°C under 5% CO$_2$. For stable HEK293 cell generation, 500 μg/ml of geneticin (G418, Life Technologies) was used.

For cell doubling assays, primary MEFs infected at a MOI of 150 with adenovirus GFP or Cre at P1 were plated at $2 \times 10^5$ in a 35-mm dish or $5 \times 10^5$ in a 60-mm dish and replated every 3–5 days. Cell numbers were determined by trypan blue exclusion assays.

## Chemical genetic kinase assay in cells

HEK293 cells stably expressing WT-S6K1 or AS-S6K1 ($4 \times 10^5$ cells/well of a 6-well plate) were cultured overnight and transfected with 2 μg of plasmid expressing the candidate substrate using the calcium phosphate method. U2OS cells ($5 \times 10^5$ cells/well of a 6-well plate) were transduced with adenoviruses at 25 MOI for WT-S6K1 or 125 MOI for AS-S6K1. Twenty-four hours after infection, cells were transfected with 5 μg of plasmid expressing the candidate substrate using Lipofectamine 2000 (Invitrogen) transfection reagent.

Forty-eight hours after transfection, cells were washed twice in PBS and incubated in labelling buffer [20 mM HEPES (pH 7.3), 100 mM KOAc, 5 mM NaOAc, 2 mM MgOAc, 10 mM MgCl$_2$, 1 mM EGTA, 0.5 mM DTT, 5 mM creatine phosphate, 57 μg/ml creatine kinase, 30 μg/ml digitonin, 100 μM N$^6$-benzyl-ATPγS (Biolog), 100 μM ATP, 5 mM GTP, 1× complete protease inhibitors EDTA-free cocktail (Roche) and 1× phosphatase inhibitor cocktail (PhosSTOP, Roche)] at room temperature for 20 min with gentle shaking. Alkylation buffer (100 mM Tris pH 8.0, 300 mM NaCl, 2% NP-40, 0.2% SDS, 20 mM EDTA, 5% DMSO, 2 mM PNBM) was added subsequently, and the reaction was allowed to continue for 1 h at room temperature with shaking. The cells were collected and the lysates cleared by centrifugation at 13,500 *g* for 10 min at 4°C followed by immunoprecipitation of the target protein.

## *In vitro* kinase assays

S6K *in vitro* kinase assays were performed according to the manufacturer's protocol of the active p70 S6 kinase (Millipore, 14-486). Briefly, HEK293 cells were transfected with Flag-tagged wild-type or Ser47Ala mutant form of ZRF1. Twenty-four hours after transfection, cells were starved overnight and treated for 3 h with Torin 1 (100 nM). Flag-tagged ZRF1 WT or Ser47Ala mutant was immunoprecipitated from 200 μg cellular lysate with anti-Flag antibody. Beads from immunoprecipitations were washed three times in lysis buffer and once in kinase buffer [4 mM MOPS (pH 7.2), 0.2 mM EDTA and 10 mM MgCl$_2$] and then used as substrates. A 53-ng weight of the p70 S6K active recombinant protein and 200 μM of ATP were added in 25 μl kinase buffer and incubated for 20 min at 30°C with gentle shaking. Reactions were stopped by the addition of SDS sample buffer.

## Western blots and immunoprecipitation

Cells were rinsed twice with ice-cold PBS and lysed in ice-cold lysis buffer (20 mM Tris–HCl (pH 8.0), 138 mM NaCl, 2.7 mM KCl, 5 mM EDTA, 20 mM NaF, 5% glycerol, 1% NP-40) in the presence of complete protease inhibitors cocktail tablets (Roche) and phosphatase inhibitor cocktail (PhosSTOP). For protein extraction from tissues, a piece of frozen tissue was ground to powder under liquid N$_2$ and lysed in ice-cold lysis buffer (20 mM Tris–HCl (pH 8.0), 138 mM NaCl, 2.7 mM KCl, 5 mM EDTA, 20 mM NaF, 5% glycerol, 1% NP-40) in the presence of complete protease inhibitors cocktail tablets (Roche) and phosphatase inhibitor cocktail (PhosSTOP). To remove cell debris, homogenates were spun at 13,500 *g* for 10 min at 4°C. Nuclear extracts were prepared using the NE-PER Kit (Pierce), according to manufacturer's recommendations.

For immunoprecipitation, 500 μg of protein extracts was incubated with anti-Flag antibody (Sigma-Aldrich) overnight at 4°C on a rotating wheel. Immunoprecipitated proteins were recovered by incubation with pre-washed protein G Sepharose beads (Amersham) for 1 h at 4°C on the wheel. After three washes in lysis buffer, the immunocomplexes were released by addition of SDS sample buffer.

Samples were resolved by SDS–PAGE before transfer onto nitrocellulose membrane (Millipore) followed by incubation with primary antibody and horseradish peroxidase-linked secondary antibodies.

## Senescence-associated β-galactosidase staining

SA β-galactosidase staining was performed using the Senescence β-Galactosidase Staining Kit (Cell Signaling, 9860) according to manufacturer's recommendations. Briefly, the cells were fixed at room temperature, for 10 min, with a solution containing 2% formaldehyde and 0.2% glutaraldehyde in PBS, washed three times with PBS and incubated for 6 h at 37°C with the staining solution containing X-gal in *N,N*-dimethylformamide (pH 6.0). Quantification was performed by counting the number of stained cells over the total number of cells in each condition. A minimum of 300 cells per condition has been counted.

## RT–qPCR

Total RNA was extracted from cells using the RNAeasy Mini Kit (QIAGEN), following manufacturer's instructions. Single-strand cDNA was synthesized from 1 μg of total RNA with SuperScript II (Invitrogen) and 125 ng random hexamer primers. Real-time quantitative PCR was performed on MX3005P instrument (Agilent) using a Brilliant III SYBR Green QPCR Master Mix (Agilent). Relative amounts of the indicated mRNAs were determined by means of the $2^{-\Delta\Delta C_T}$ method, normalizing with β-actin levels.

## Immunofluorescence

Cells were grown on a 13-mm-diameter coverslip and fixed in 4% PFA for 10 min at RT. After 3 PBS washes, cells were permeabilized with 0.3% Triton X-100 in PBS and blocked in 3% BSA 3% normal goat serum in PBS. Samples were then incubated with primary antibody (anti-γH2AX 1/200; anti-53BP1 1/100) overnight at 4°C, washed three times with PBS and incubated an additional hour at room temperature with Alexa568-conjugated secondary antibody (Molecular Probes). Coverslips were finally mounted on slides with mounting medium containing DAPI (Vector Laboratories).

## Animals

*Tsc1*$^{fl/fl}$ mice were obtained from The Jackson Laboratory. Generation of *S6K1*$^{-/-}$*S6K2*$^{-/-}$-deficient mice (C56BL/6-129/Ola) has been previously described. All mice used in this study were on mixed genetic background (C57BL/6J, BALB/cJ or 129/SvJae). Each genotype was compared with wild-type mice of the same genetic background. Mice were maintained at 22°C with a 12-h dark/12-h light cycle and fed a standard chow diet (Teklad global protein diet; 20% protein, 75% carbohydrate and 5% fat). All studies were done in 3-month-old male animals and were approved by the Direction Départementale des Services Vétérinaires, Préfecture de Police, Paris, France (authorization number 75-1313) and the ethical committee of Paris Descartes University.

## Statistical analysis

Data are shown as means ± SEM. Analysis were performed by unpaired, two-tailed, Student's *t*-test or two-way ANOVA. $P < 0.05$ was considered statistically significant.

**Expanded View** for this article is available online.

## Acknowledgements

We are grateful to the members of INSERM-U1151 for support; to Manuel Serrano, Luciano Di Croce and David Kwiatkwoski for sharing reagents; and to Stefano Fumagalli for critically reading the manuscript. We thank members of the VVTG platform (SFR Necker, France) for the technical help and Estelle Le Borgne and Jean Pierre Laigneau for the graphical assistance. G.B. received a fellowship from the Swiss National Science Foundation and from Association pour la Recherche sur le Cancer. This work was supported by grants from the European Research Council, Fondation Schlumberger pour l'Education et la Recherche, Agence Nationale de la Recherche, Association Sclerose Tubereuse de Bourneville, Inserm AgeMed transversal program, Institut National du Cancer to M.P.

## Author contributions

MB, GB and CT performed and interpreted most experiments, and were involved in designing some experiments. VK carried out experiments and performed kinase assays. DDD helped preparing primary cultures and took care of the animal colonies. SF produced all viral preparations. MP conceived and supervised the project, designed experiments, interpreted data, and wrote the manuscript. All authors commented on the manuscript.

## Conflict of interest

The authors declare that they have no conflict of interest.

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
