## [Review Process File · The EMBO Journal]

Manuscript EMBO-2016-94966

ZRF1 is a novel S6 kinase substrate that drives the senescence programme

Manuela Barilari, Gregory Bonfils, Caroline Treins, Vonda Koka, Delphine De Villeneuve, Sylvie Fabrega and Mario Pende

Corresponding author: Mario Pende, INSERM

Review timeline:

Submission date:	07 June 2016
Editorial Decision:	12 August 2016
Revision received:	14 December 2016
Editorial Decision:	11 January 2017
Revision received:	20 January 2017
Accepted:	23 January 2017

Editor: Andrea Leibfried

Transaction Report:

1st Editorial Decision

12 August 2016

Thank you for submitting your manuscript to The EMBO Journal. Andrea Leibfried is the handling editor on the manuscript but as she is away at the moment I have stepped in as 2nd editor on the manuscript.

Your manuscript has now been seen by three referees and their comments are provided below. As you can see there is an interest in the study, but the referees also find that some further analysis is needed to consider publication here. Additional mechanistic insight into how S6K-dependent ZRF1 phosphorylation affects p16 expression and senescence would be good. Also further support for that ZRF1 is a S6K substrate is also needed. Given the referees' positive recommendations, I would like to invite you to submit a revised version of the manuscript, addressing the comments of all three reviewers. I should add that it is EMBO Journal policy to allow only a single major revision and that it is important to resolve the major concerns raised at this stage.

REFeree REPORTS

Referee #1:

The paper by Bonfils et al identifies the ZRF1 protein as a novel substrate of the S6K1/2 kinase that is required for induction of senescence downstream of oncogenic mTORC1 signaling. The authors

demonstrate that, in TSC1-deleted cells, both S6K and ZRF1 are mediators of a senescence cascade that culminates in the transcriptional induction of p16. Moreover, through a chemical genetic screen with a gatekeeper S6K mutant, they provide strong evidence that ZRF1 is a direct substrate of S6K. The data presented are of very good quality and the controls are rigorous. The paper provides insight into an important but poorly understood aspect of mTORC1 signaling, namely the relationship between aberrant mTORC1 and induction of senescence. However, some aspects require further clarification.

1. TSC1 deletion results in transcriptional upregulation of p16 that requires S6K1/2. Can inducible p16 expression restore senescence in TSC1-S6K1/2 knock out cells?
2. In Fig. 7C, p16 protein levels correlate with ZRF1 status. Is this the case also for p16 mRNA levels? Also, can p16 restore senescence in ZRF1-deleted cells?
3. Although the phosphorylation data using AS-S6K are compelling, the gold standard to conclusively establish that ZRF1 is an S6K substrate is an in vitro kinase assay with purified proteins. Such experiment would considerably strengthen the paper.
4. Additional mechanistic insight is required into how ZRF1 promotes p16 expression downstream of S6K1/2. Is nuclear ZRF1 sufficient to induce senescence? The authors could test that by expressing an NLS-tagged ZRF1. Also, does ZRF1 regulate p16 expression via its interaction with Id-family transcription factors?

Referee #2:

In this manuscript, Bonfils et al employed a chemical genetic screen system adapted for S6K to examine potential substrates of S6K. Among 10 candidates examined, 3 proteins were found to be potential S6K substrates. One of these proteins ZRF1 was chosen to validate by examining the phosphorylation of ZRF1 in the presence and absence of S6K1/2. Ectopic ZRF1 was shown to restore the senescence in TSC deficient cells whereas its phosphor-mutant abolished this capacity.

The initial finding is interesting and the authors established substantial evidence that ZRF1 is a substrate of S6K1/2. However, the role for senescence of ZRF1 was quite superficial and no mechanistic data were provided to explain the connection between phosphorylation of ZRF1 and senescence, though p16 was shown to be associated with this process. Does loss of p16 rescue the senescence mediated by ZRF1 phosphorylation? Does p53/p21 play any role in ZRF1-mediated senescence. In addition, many data lack statistical quantification especially in the Western blotting analyses. The internal control for cytosolic fraction in Figure 7 is missing. The details on the passages of the cells used in the studies are often missing (Fig 2 A,B). This is very important information as the difference was only shown to be significant at later passages (Fig 1 C and D). It is possible that the difference was not significant at early passage (Fig 2). This must be clarified. In Fig 7B, shZRF1 significantly reduced senescence after TSC1 is lost. Is the senescence delayed or diminished? What happen to cells that deficient for ZRF1 in the presence and absence of TSC1? The role of S6K1 and S6K2 on ZRF1 phosphorylation is unclear. Authors claimed that S6K1 and S6K2 are compensatory and deletion of both is required for diminishing phosphorylation of ZRF1. However, there are no data to support this claim. On the other hand, do the two S6K have similar preference and comparable activities on the phosphorylation at S47 and S49 of ZRF1? Although some data presented suggesting that pZRF1 is important in senescence, there is little evidence that it is the driving force of senescence. It is part of the signaling cascade.

Referee #3:

This manuscript describes a senescence response elicited by constitutive mTOR activation and its dependence on S6 kinase-mediated phosphorylation of ZRF1, an epigenetic transcriptional regulator and ribosomal chaperone.

Overall, this is an interesting manuscript that adds to our understanding of how mTOR regulates cellular physiology. In addition, the authors make excellent use of a relatively recent method for identifying kinase substrates to identify ZRF1 as a new S6K1 substrate. Some suggestions for improving the manuscript's impact and clarity are below.

1) Floxed Tsc1 cells treated with Cre show a modest induction of SA-beta-Gal (fig 1B), p53 protein (fig 1A), albeit a more robust induction of p16Ink4a protein (fig 1A) and increase in doubling time (fig 1c)? But are the cells really senescent? Certainly not all of them are (30% SA-beta-Gal +). I would urge the authors to explore additional senescence markers, and also suggest they consider that p16Ink4a is more promiscuous in MEFs compared to human fibroblasts, and expression of this protein is not always correlated with arrested growth (often due to upregulation of Cdk4 or 6).

2) Figure 6 used 'immortalized cells' (presumably MEFs) - were these cells spontaneously immortalized or were they immortalized by an oncogene such as T antigen? Why were immortalized cells used for these experiments, whereas presumably the other experiments used primary cells?

3) Figures 5 and 6 also show that S6K1 and S6K2 are at least partially redundant for ZRF1 phosphorylation on serine 47. Is it possible there are other S6K1 or S6K2 phosphorylation sites that contribute to the senescence-regulating activity of ZRF1? If this is known, the authors should comment on the possibility - if it is not known, they should consider the possibility in their discussion.

4) One weakness of the paper is the assumed mechanistic links between S6K-dependent ZRF1 phosphorylation, p16Ink4a induction and senescence are somewhat tenuous. As noted above, the argument for senescence is partial at best (going from roughly 15% to <30% SA-beta-Gal positivity in Fig 7). The correlation with p16Ink4a expression is stronger, but the presumed mechanism by which ZRF1 induces p16Ink4a is by displacing polycomb-repressive complexes (which are known to repress the Ink4a/Arf locus), as the authors suggest in the Discussion. Is there any evidence for this mechanism?

Minor comments:

- 1) The manuscript contains a few grammatical errors and would benefit from careful editing.
- 2) In the introduction, the authors argue that S6K1 is a more important target of mTOR than S6K2, based on physiological differences in certain tissues from S6K1- vs S6K2-deficient mice. It should be noted, however, that the (modestly) extended life span of S6K1-deficient mice was gender-specific (females only).

1st Revision - authors' response

14 December 2016

Report continued on next page.

We thank the three reviewers for their comments and we appreciate that they found broad interests in our work. We think we largely addressed the reviewer's comments and improved the manuscript. In this revised version, we provide new functional data delineating the pathway from S6 kinases to ZRF1 and p16 in the control of senescence, downstream of TSC1 mutations and mTORC1 activation. Briefly, we address three main points. First, we provide further analysis demonstrating that ZRF1 is a direct S6K substrate by *in vitro* kinase assays. Second, we clarify that both S6K1 and S6K2 concur in ZRF1 phosphorylation, though their relative activities are distinct. Third, we strengthen the connection between S6K activity, p16 expression and senescence by using pharmacological tools as well as adenoviral transduction of p16 cDNA and shRNA. Finally, we include a thorough panel of quantifications and controls to interpret the data on genetic and pharmacological treatments, as requested by the reviewers. There is a large amount of new data that are included in the 7 main figures and 2 expanded view figures of this revised manuscript.

Below is our response to the reviewer comments.

Referee #1:

The paper by Bonfils et al identifies the ZRF1 protein as a novel substrate of the S6K1/2 kinase that is required for induction of senescence downstream of oncogenic mTORC1 signaling. The authors demonstrate that, in TSC1-deleted cells, both S6K and ZRF1 are mediators of a senescence cascade that culminates in the transcriptional induction of p16. Moreover, through a chemical genetic screen with a gatekeeper S6K mutant, they provide strong evidence that ZRF1 is a direct substrate of S6K. The data presented are of very good quality and the controls are rigorous. The paper provides insight into an important but poorly understood aspect of mTORC1 signaling, namely the relationship between aberrant mTORC1 and induction of senescence. However, some aspects require further clarification.

1. TSC1 deletion results in transcriptional upregulation of p16 that requires S6K1/2. Can inducible p16 expression restore senescence in TSC1-S6K1/2 knock out cells?

We thank the reviewer for the comments on the quality, rigor and importance of our analysis. This reviewer suggests p16 gain-of-function approaches to clarify the interaction between S6Ks and p16 on the control of senescence. Reviewer #2 would also like to see loss-of-function approaches. In this revised version, we include adenoviral transduction of p16 cDNA and shRNA. In the new Figure 3A, we show that shRNA against p16 decreases senescence of TSC1 mutant cells, as assessed by senescence associated beta galactosidase staining. In addition, p16 overexpression rescues the senescence program of cells treated with the specific S6K inhibitor PF-4708671. We thank both reviewers because we think these new data considerably strengthen the positive relationship between S6K activity and p16 expression in the control of TSC1 mutant cell senescence.

2. In Fig. 7C, p16 protein levels correlate with ZRF status. Is this the case also for p16 mRNA levels? Also, can p16 restore senescence in ZRF1-deleted cells?

In this revised version, we show in the new Figures 1, 3 and 7 that S6K activity controls p16 mRNA levels and this correlates with the status of ZRF1 phosphorylation. Both S6K pharmacological inhibition and genetic deletion decrease p16 mRNA levels, suggesting a regulation at the level of transcription. The rescue of the senescence program by adenoviral-mediated overexpression of p16 in ZRF1 knock-down cells is a complicated experiment to perform and interpret. The cells would need to be treated first with adenoviral transduction of Cre to delete the TSC1 gene, followed by lentiviral transduction of ZRF1 shRNA, followed by adenoviral transduction of p16. The kinetics and transduction efficiency are difficult to control in three distinct passages and infection protocols. We recently generated knock-out and phospho-mutant knock-in ZRF1 mutant mice by CRISPR-Cas9 technology. The mice are at the F1 stage and need to be bred and expanded. We think that in the next two years, this strategy would provide the best experimental set-up for the proposed experiments.

3. Although the phosphorylation data using AS-S6K are compelling, the gold standard to conclusively establish that ZRF1 is an S6K substrate is an *in vitro* kinase assay with purified proteins. Such experiment would considerably strengthen the paper.

This is an important point that led us to the experiment in the new Figure 5C. We immunopurified unphosphorylated ZRF1 and performed *in vitro* kinase reactions using recombinant S6K1. The data clearly show that recombinant S6K1 is able to directly phosphorylate ZRF1 Ser 47 and that the phospho-specific antibody exclusively recognizes this site.

4. Additional mechanistic insight is required into how ZRF1 promotes p16 expression downstream of S6K1/2. Is nuclear ZRF1 sufficient to induce senescence? The authors could test that by expressing an NLS-tagged ZRF1. Also, does ZRF1 regulate p16 expression via its interaction with Id-family transcription factors?

In this study, we have limited our analysis to the reported ZRF1 interactors that have been shown to control senescence, i.e. the PRC1 complex and Id1 proteins. Unfortunately, in our cellular system we failed to reveal a functional role of ZRF1 phosphorylation with these putative partners. In HEK-293 cells, we were able to detect the interaction between overexpressed ZRF1 and ID1 (Figure below), though we failed to detect it in the MEFs system used for the senescence studies, possibly due to sensitivity issues. In both HEK-293 cells and MEFs, we showed that overexpressed ZRF1 binds to the chromatin (Figure below). However, in these cell types we were not able to reproduce the published evidence that ZRF1 can displace the BMI1- and Ring1B- containing PRC1 complex. Future studies should use unbiased approaches to compare the interactome of wild type and phospho-mutant ZRF1. We plan to do this analysis in the ZRF1 knock-out and knock-in primary MEFs that should be generated in the future.

Figure : Interaction of ZRF1 with ID1 and PRC1

- A- HEK-293T cells were transfected with HA-tag ID1 and flag-tag wt ZRF1 or mutant ZRF1^{S47A}. After immunoprecipitation with HA antibody, the co-immunoprecipitation was revealed by ZRF1 antibody.
- B- HEK-293T cells were transfected with the flag tagged wt ZRF1 or the mutant ZRF1^{S47A}. Forty eight hours post transfection cells were collected, nuclear and cytoplasmic fractions were extracted and analysed by western blot using the indicated antibodies.
- C- Primary MEF were transduced with adenoviruses expressing wt ZRF1 and the mutant ZRF1^{S47A}. Forty eight hours post infection cells were collected, nuclear and cytoplasmic fractions were extracted, and analysed by western blot using the indicated antibodies.

Referee #2:

In this manuscript, Bonfils et al employed a chemical genetic screen system adapted for S6K to examine potential substrates of S6K. Among 10 candidates examined, 3 proteins were found to be potential S6K substrates. One of these proteins ZRF1 was chosen to validate by examining the phosphorylation of ZRF1 in the presence and absence of S6K1/2. Ectopic ZRF1 was shown to restore the senescence in TSC deficient cells whereas its phospho-mutant abolished this capacity.

1-The initial finding is interesting and the authors established substantial evidence that ZRF1 is a substrate of S6K1/2. However, the role for senescence of ZRF1 was quite superficial and no mechanistic data were provided to explain the connection between phosphorylation of ZRF1 and senescence, though p16 was shown to be associated with this process. Does loss of p16 rescue the senescence mediated by ZRF1 phosphorylation? Does p53/p21 play any role in ZRF1-mediated senescence.

We thank both reviewers #2 and #1 for raising this important point that led to the p16 gain-of-function and loss-of-function approaches shown in the new Figure 3. As detailed in our answer to the point #1 of reviewer #1, we think that our new data strengthen the connection between S6K activity and p16 expression in the control TSC1-mutant cell senescence.

2-In addition, many data lack statistical quantification especially in the Western blotting analyses.

In this version, we added quantification and statistical analysis in the new Figures 1, 3B, 3C, 7C, Expanded View Figure 2.

3-The internal control for cytosolic fraction in Figure 7 is missing.

We added tubulin as a cytosolic marker in the new Figure 7A.

4-The details on the passages of the cells used in the studies are often missing (Fig 2 A,B).

This is very important information as the difference was only shown to be significant at later passages (Fig 1 C and D). It is possible that the difference was not significant at early passage (Fig 2). This must be clarified.

We are sorry that these important issues were not clearly stated in the previous version. The DNA damage was assessed at late passages, when TSC1 mutant cells started to senescence and DNA damage would be more likely to accumulate. As

shown in Figure 2, at passage 7 we could not detect differences among the genotypes, using two distinct markers of the DNA damage response.

5-In Fig 7B, shZRF1 significantly reduced senescence after TSC1 is lost. Is the senescence delayed or diminished? What happen to cells that deficient for ZRF1 in the presence and absence of TSC1?

As shown in Figure 7B, there is a tendency of shZRF1 to also decrease the senescence program in cells carrying wild type TSC1. However this effect was less prominent than in TSC1 mutant cells at passage 7 and did not reach statistical significance. As shown in Figure 1C and clarified in the text of this revised version, the inactivation of the S6K pathway does not cause a complete block of the senescence program, but a delay in the kinetics.

6-The role of S6K1 and S6K2 on ZRF1 phosphorylation is unclear. Authors claimed that S6K1 and S6K2 are compensatory and deletion of both is required for diminishing phosphorylation of ZRF1. However, there are no data to support this claim. On the other hand, do the two S6K have similar preference and comparable activities on the phosphorylation at S47 and S49 of ZRF1?

We thank the reviewer for this important point that we addressed in the new Figure 6C. In mouse livers after refeeding, the combined deletion of both S6K1 and S6K2 are required to abrogate ZRF1 phosphorylation. The single deletion of S6K1 or S6K2 results in a residual ZRF1 phosphorylation, indicating that each kinase can partly compensate for the loss of the other. It is interesting that S6K1 and S6K2 appear to have relatively distinct activities towards rpS6 and ZRF1. We confirm many published studies indicating that rpS6 is a preferential substrate for S6K2. In contrast, here we demonstrate that S6K1 deletion has more potent effects than S6K2 on phospho-ZRF1 vs phospho-rpS6.

7-Although some data presented suggesting that pZRF1 is important in senescence, there is little evidence that it is the driving force of senescence. It is part of the signaling cascade.

We completely agree with the reviewer and clarified in the discussion that S6Ks turn on a senescence program by affecting multiple substrates. In this study, we also provide evidence that lamin A may be an S6K substrate, and others potentially involved in senescence may be discovered in the future.

Referee #3

This manuscript describes a senescence response elicited by constitutive mTOR activation and its dependence on S6 kinase-mediated phosphorylation of ZRF1, an epigenetic transcriptional regulator and ribosomal chaperone.

Overall, this is an interesting manuscript that adds to our understanding of how mTOR regulates cellular physiology. In addition, the authors make excellent use of a relatively recent method for identifying kinase substrates to identify ZRF1 as a new S6K1 substrate. Some suggestions for improving the manuscript's impact and clarity are below.

1) Floxed Tsc1 cells treated with Cre show a modest induction of SA-beta-Gal (fig 1B), p53 protein (fig 1A), albeit a more robust induction of p16Ink4a protein (fig 1A) and increase in doubling time (fig 1c)? But are the cells really senescent? Certainly not all of them are (30% SA-beta-Gal +). I would urge the authors to explore additional senescence markers, and also suggest they consider that p16Ink4a is more promiscuous in MEFs compared to human fibroblasts, and expression of this protein is not always correlated with arrested growth (often due to upregulation of Cdk4 or 6).

We thank this reviewer who finds the methodology to discover new S6K substrates excellent, and the new evidence of the role in the control of senescence interesting. As shown in Figure 1C and clarified in the text, S6K inactivation delays, rather than blocking the senescence program. In this study we focus our analysis at passages 5-7 when TSC1 mutant cells start to enter the senescent program while the combined

deletion of S6K delays the kinetics. We also used Rheb overexpression to induce senescence, as an alternative strategy to the TSC1 deletion. As shown in the Figure below, S6K inactivation is equally effective in counteracting this trigger of senescence. In our analysis, we used the expression of p16, p21, p53 and the doubling time in addition to the beta galactosidase assay to measure senescence. We know that Jesus Gil and Dominic Withers (Imperial College of London) have complementary evidence of S6K controlling the senescence associated secretory phenotype (SASP). Since we are collaborating with them on this aspect, we would prefer not to include these data on the present manuscript.

Figure : S6 kinases control senescence induced by Rheb overexpression

Quantification of SA-β-galactosidase staining performed on primary MEF *TSC1^{fl/fl}* or *TSC1^{fl/fl};S6K1^{-/-};S6K2^{-/-}* that were transduced with lentiviruses expressing FLAG-Rheb or EGFP at passage P3. Thirty six hours post transduction the cells were selected with puromycin. At passage P7 cells were fixed and subjected to SA-β-galactosidase staining.

2) Figure 6 used 'immortalized cells' (presumably MEFs) - were these cells spontaneously immortalized or were they immortalized by an oncogene such as T antigen? Why were immortalized cells used for these experiments, whereas presumably the other experiments used primary cells?

We are sorry if this was confusing in the previous version. We wanted to use different cell types and conditions to corroborate our discovery that ZRF1 is a novel substrate

of S6Ks, even if not all of them are relevant to the study of senescence. Hence we used HEK-293, U2OS cells, liver, muscle, fat tissues, immortalized TSC2 mutant MEFs and primary TSC1 mutant MEFs. As the reviewer correctly states, the primary TSC1 mutant MEFs are directly relevant for the senescent phenotype. The ZRF1 phosphorylation in these cells is assessed in the new Figure 3 after treatment with a pharmacological S6K inhibitor, and in the Figure 6E by combining the genetic deletion of S6Ks.

3) Figures 5 and 6 also show that S6K1 and S6K2 are at least partially redundant for ZRF1 phosphorylation on serine 47. Is it possible there are other S6K1 or S6K2 phosphorylation sites that contribute to the senescence-regulating activity of ZRF1? If this is known, the authors should comment on the possibility - if it is not known, they should consider the possibility in their discussion.

This is an important point that was also partly raised by reviewer #2, point #6 (see above). In the revised version we included the comparison between S6K1 and S6K2, showing the redundancy but also the relative kinase activities (new Figure 6C). In the experiment with the analogue sensitive kinase (Figure 5B), we showed that the mutation of Ser47 is sufficient to abrogate ZRF1 thiophosphorylation by AS-S6K1. In addition, in the new Figure 5C using recombinant S6K1 we show that the kinase can directly phosphorylate this site *in vitro*. For all these reasons, we consider unlikely the presence of additional S6K phosphorylation sites in the ZRF1 protein.

4) One weakness of the paper is the assumed mechanistic links between S6K-dependent ZRF1 phosphorylation, p16Ink4a induction and senescence are somewhat tenuous. As noted above, the argument for senescence is partial at best (going from roughly 15% to <30% SA-beta-Gal positivity in Fig 7). The correlation with p16Ink4a expression is stronger, but the presumed mechanism by which ZRF1 induces p16Ink4a is by displacing polycomb-repressive complexes (which are known to repress the Ink4a/Arf locus), as the authors suggest in the Discussion. Is there any evidence for this mechanism?

Following the suggestions of the three reviewers, in this revised version we strengthen the relationship between S6Ks and p16 in the control of senescence using a variety of pharmacological tools and viral vectors (see point #1, reviewer #1). We think that our study has been considerably improved on this aspect and opens new avenues of investigation on the control of p16 expression by the mTORC1 pathway. As detailed in the answer to the point #4 of the reviewer #1, we tried several experiments to assay the interaction between ZRF1, ID1 and the displacement of the PRC1 complex. Since we failed, possibly due to technical limitations, in the future we should try unbiased interactome approaches using the new ZRF1 mouse lines that are being generated by CRISPR-Cas 9 technology.

Minor comments:

1) *The manuscript contains a few grammatical errors and would benefit from careful editing.*

The revised manuscript has been edited for grammatical errors.

2) *In the introduction, the authors argue that S6K1 is a more important target of mTOR than S6K2, based on physiological differences in certain tissues from S6K1- vs S6K2-deficient mice. It should be noted, however, that the (modestly) extended life span of S6K1-deficient mice was gender-specific (females only).*

We agree with the reviewer. In this revised version we clarify that our data are relevant for the senescent program triggered by TSC1 mutations and mTORC1 overactivation. Whether this is also relevant for physiological ageing will be the scope of future studies using a cohort of old S6K and ZRF1 mutants.

Thank you for submitting your revised manuscript for our consideration. Please excuse the delay in getting back to you, but I have now heard back from all referees. I am happy to inform you that they are all in favor of publication of your revised manuscript. I am thus happy to accept your manuscript in principle for publication here.

REFeree REPORTS

Referee #1:

In their revised manuscript Barilari, Bonfils, Treins et al have satisfactorily addressed the concerns raised in the original submission. In particular, evidence is provided that S6-kinase 1 and 2 phosphorylate ZRF1 both in cells and in vitro.

The authors also clarify the relationship between S6K1 and p16 with two new experiments. First, they provide evidence that p16 overexpression rescues the loss of senescent program caused by S6K inhibition. Second, they provide new evidence that S6K1 controls p16 expression at the transcriptional level via ZRF1.

How ZRF1 phosphorylation by S6K leads to upregulation of p16 expression remains an unexplained but important point, which should be addressed in future work.

Referee #2:

In the revised manuscript, authors have provided additional data and statistical analyses on gene expression, etc. They have provided new supporting experimental data to address the issues concerned and toned down their claims. The revised version is significantly improved. I am satisfied with their response to the questions raised and recommend acceptance.

Referee #3:

In their revised manuscript, the authors have answered all of the major comments made by the reviewers. Some minor comments, or comments that will require extensive additional experimentation remain, but these deficits are offset by the significance of the findings and the many additional experiments in the revised version that strengthen its conclusions.

Corresponding Author Name: Mario Pende
Journal Submitted to: EMBO Journal
Manuscript Number: EMBOJ-2016-94966R